# Modification of Biomass-Derived Nanoporous Carbon with Nickel Oxide Nanoparticles for Supercapacitor Application

Bakhytzhan Lesbayev [1,2], Moldir Auyelkhankyzy [1,2,*], Gaukhar Ustayeva [1,2], Mukhtar Yeleuov [2], Nurgali Rakhymzhan [2], Yerkebulan Maral [1,2] and Aidos Tolynbekov [2]

1 Faculty of Chemistry and Chemical Technology, al-Farabi Kazakh National University, Almaty 050040, Kazakhstan
2 Laboratory of Synthesis of Carbon Nanomaterials in Flames, Institute of Combustion Problems, Almaty 050012, Kazakhstan
* Correspondence: moldir.auyelkhankyzy@kaznu.edu.kz

**Abstract:** Supercapacitors are one of the promising devices for the accumulation and storage of electrical energy. The purpose of this study is to develop a synthesis and modification method of carbon material to improve the electrochemical characteristics of a supercapacitor. In the proposed study, by varying the sequence and parameters of the processes of carbonization, mechanoactivation and thermochemical activation, the conditions for obtaining nanoporous carbon with a specific surface area of 2200 ($\pm$50) m$^2$/g from walnut shells (WSs) are optimized. In addition, to increase the electrochemical efficiency of the electrode material, the resulting nanoporous carbon was modified with nickel oxide (NiO) nanoparticles by the thermochemical method. It is shown that the modification with nickel oxide nanoparticles makes it possible to increase the specific capacitance of the supercapacitor electrode by 16% compared to the original unmodified nanoporous carbon material.

**Keywords:** biomass-derived; walnut shell; activated carbon; modification; CVD; supercapacitors

## 1. Introduction

Batteries (galvanic cells), capacitors, and supercapacitors are currently used as electrical systems for the accumulation and storage of electrical energy. Supercapacitors, in their electrochemical characteristics, occupy an intermediate position between batteries and capacitors, but at the same time, they have a high specific power, a record charge and discharge rate, and an almost unlimited service life [1]. Due to these advantages, they are currently the most promising devices for the accumulation and storage of electrical energy [2]. The paper by Şahin M et al. [3] provides a comprehensive review of the literature on supercapacitor designs, operating principles, specifications, and classifications in a comparable manner, as well as a feasibility study on the use of supercapacitors in applications such as transportation, electric vehicles, hybrid power systems, and military applications.

In recent decades, porous carbon materials obtained from biomass have become widely used as electrode materials for supercapacitors due to their low cost, renewability, rich reserves, environmental friendliness, and unique properties (such as large surface area) [4]. Biomass as a feedstock for obtaining porous carbon materials is also attractive because the problems associated with the disposal and use of plant raw materials waste remain relevant due to environmental and economic factors [5,6]. The application of biomass carbon electrode materials for the development of various types of supercapacitors with enhanced electrochemical performance, high-rate capability, and cycling stability is discussed in [7–11]. Rice, walnuts, etc., are widely grown worldwide. In the process of processing them, many different wastes are generated, from which valuable nanoporous carbon materials are obtained by carbonization and activation processes. Comprehensive studies have been carried out to obtain carbon materials based on rice husks (RHs) with various types of structure, such as hierarchical [12], honeycomb [13], etc. [14–16], which

were used as electrode materials for supercapacitors. However, there are only a few articles on walnut shell-based carbon for supercapacitor electrode applications. Ahmed et al. [17] prepared activated charcoal with a high surface area of 3800–4320 $m^2$/g using Iraqi walnut shells as a carbon source. The obtained material shows excellent physical and adsorption properties and can be applied to the removal of heavy elements during water purification. Zhang et al. [18] fabricated a WS@Ni-MOF/SPANI composite material and used it as new electrode material for high-performance supercapacitors. WS@Ni-MOF/SPANI electrode material showed excellent electrochemical properties, such as a specific capacitance was 1722 F/g at 1 A/g. Fu et al. [19] synthesized WS-derived hierarchical porous carbon with a specific surface area of 1037.31 $m^2$/g and applied it as a supercapacitor electrode material. The sample exhibited good electrochemical performances, including a specific capacitance of 262.74 F/g at 0.5 A/g and a rate capability of 224.60 F/g at even 10 A/g. Xu et al. [20] prepared WS-derived porous carbon material using a hydrothermal method and used it as the electrode material in a symmetric all-solid-state supercapacitor. The supercapacitor demonstrated good electrochemical performance with high specific capacitance and good cycle stability with only a slight capacitance loss of less than 10%. Nevertheless, the further development of new electrode materials with porous structures, high performance, good stability, and adjustable chemical properties is highly required for the development of next-generation supercapacitors. Researchers all over the world seek and study various types of synthesis, doping [21], and modification [22,23] methods to obtain advanced carbon materials with improved physicochemical properties for high-performance supercapacitors. One of the effective methods to prepare such electrode materials is incorporating pseudocapacitive materials into porous carbons. Among various pseudocapacitive materials, transition metal oxides, such as NiO, $MnO_x$, and $Fe_2O_3$, generally possess high energy density and large capacitance and are easy to be converted into nanoparticles for incorporation on the surface of the carbon materials.

Based on this, the combination of NiO nanoparticles with cheap activated carbon materials may be a promising approach to preparing high-performance and low-cost carbon-based electrode materials. In this work, a mechanical activation method by a high-energy 3D ball mill was used to increase the effective specific surface area of a porous carbon obtained from WSs. Then, the obtained mesoporous carbon was modified with nickel oxide nanoparticles in order to increase the specific capacitance of supercapacitor electrodes.

## 2. Experimental Part

### 2.1. Preparation of WS-dAC

The WSs obtained from a local farm in Kazakhstan were used as a carbon precursor in this work. Before the experiment, WSs were washed with water to remove impurities and then dried at 100 °C for 12 h. The cleaned WSs were crushed at several stages. The WS samples were degassed at 120 °C for 12 h under vacuum prior. First, the raw material was crushed with an impact mill for 2 min, and then the materials were degassed under a vacuum prior. The crushed WSs were transferred into a horizontal tube furnace and carbonized for 1 h at 550 °C. The carbonization process was protected by purging with argon. WSs mainly contain cellulose and hemicellulose, as well as lignin, like all other lignocellulosic materials. The thermal decomposition and kinetics of hemicellulose, cellulose, and lignin are well studied, and it is known that their complete decomposition occurs in the temperature ranges of ~210–325, ~310–400, and ~160–900 °C, respectively [24–27]. Previous studies have shown that the largest mass loss (56%) of walnut shells during pyrolysis occurs in the temperature range of 190–380 °C. And continuous heating of the biomass to a temperature of 800 °C showed that the mass loss began to decrease. The average residual weight was determined to be 21% at the end of the overall thermal decomposition process. As a result of our research, it was found that the weight loss of the walnut shell during carbonization is insignificant above a temperature of 550 °C, which means that the main thermal decomposition of cellulose, hemicellulose and lignin occurs below this temperature.

The obtained carbonized WS was ground in a high-energy 3D ball mill for 15 min. The mechanoactivation process initiates the intensive formation of point and linear defects (ionic and atomic vacancies, interstitial ions, and dislocations), which leads to the formation of microcracks and additional channels and expands access to remove volatile components at the stage of subsequent thermochemical activation, which ultimately contributes to the effective formation of the system meso- and micropores. Grounded carbonized WSs in a high-energy 3D ball mill was subjected to further thermochemical activation to increase its specific surface area. The potassium hydroxide (KOH) was used as an activating agent. KOH and carbonized product were mixed at a mass ratio of 4:1 (KOH:C = 4:1). The temperature of the mixture in the reactor was maintained at 100 °C for 12 h until carbonized WSs were well impregnated with KOH. Then the temperature was raised to 850 °C and heat treated at this temperature for 90 min in an argon atmosphere to activate the carbonized WSs. The activated product was washed with DI water, then with HCl to remove the d-elements (Fe, Cr, Ti, etc.), as well as with DI water until the filtrate was neutral (pH = 7). The resulting activated product was dried at room temperature for 12 h; after that, it dried at 100 °C for 8 h in a vacuum dryer to obtain the walnut shells-derived activated carbon (WS-dAC). Under the above optimized conditions, one can stably synthesize nanoporous carbon materials with a specific surface area of 2200 ($\pm$50) m$^2$/g, which have a mostly mesoporous structure.

### 2.2. Modification of WS-dAC

To improve the electrochemical characteristics of the supercapacitor, WS-dAC was modified by introducing nickel oxide nanoparticles into its structure. The method for obtaining WS-dAC modified with nickel oxide nanoparticles consisted of the following operations. The autoclave was loaded with 0.5 g of activated carbon and poured with 50 mL of an aqueous solution of nickel nitrate hexahydrate ($Ni(NO_3)_2 \cdot 6(H_2O)$). After that, the autoclave was placed in a hot oven and kept at a temperature of 200 °C for 5 h. Nickel nitrate-impregnated WS-dAC was cooled to room temperature, washed with cold distilled water, filtered, and dried at room temperature for 12 h. The dried material was additionally subjected to vacuum drying at a temperature of 100 °C for 4 h. The dried WS-dAC with nickel nitrate hexahydrate deposited in the pores was subjected to heat treatment for an hour in an argon atmosphere in a horizontal tube furnace at a temperature of 300 °C. After heat treatment in an inert atmosphere, nickel oxide nanoparticles were formed in the porous structure of WS-dAC. To control the size and concentration density of nickel nanoparticles, experiments were carried out using an aqueous solution of nickel nitrate with a concentration of 0.001 M, 0.005 M, and 0.01 M (WS-dAC/NiO-x: WS-dAC/NiO-0.001, WS-dAC/NiO-0.005, WS-dAC/NiO-0.01). Nickel oxide nanoparticles are formed according to the following reaction: $2\{Ni(NO_3)_2 \cdot 6(H_2O)\} \rightarrow 2NiO + 4NO_2 + O_2 + 12H_2O$.

### 2.3. Materials Characterizations

The morphological and structural characteristics of obtained materials were studied using a scanning electron microscope (SEM, Jeol JSM-6490LA) and transmission electron microscopy (TEM JEM-2100CX). Elemental content of the samples was studied by Energy Dispersive X-ray Analysis (EDAX). The specific surface area was determined with an Autosorb IQ automatic physisorption micropore analyzer. Electrochemical characteristics were measured by cyclic voltammetry (CV) and galvanostatic charge-discharge (GCD) methods.

### 2.4. Electrochemical Measurements

Electrochemical measurements of the obtained materials based on WSs and their modification with NiO particles (WS-dAC/NiO-0.001, WS-dAC/NiO-0.005, WS-dAC/NiO-0.01) were carried out using a potentiostat-galvanostat with an EIS measurement module (EIS (P-40X c FRA-24M)). For this, two current collectors with an electrode layer of the same mass deposited on their surface were selected to collect a two-electrode cell. 6 M KOH was used as the electrolyte, and filter paper (Cat No. 1001-110, WhatmanTM) was used

as the separator. Cyclic voltammetry (CV) and galvanostatic charge-discharge (GCD) were performed to study the electrochemical properties of the obtained materials. The specific capacitance is determined from the curve of cyclic voltammetry and galvanostatic charge-discharge [28,29].

### 3. Results

The morphological, structural and elemental characteristics of the resulting WS-dAC/NiO-x composites were studied by SEM, TEM, and EDAX analysis. The manuscript presents the results of SEM, TEM, and EDAX analysis of the WS-dAC/NiO-0.005 sample since the electrochemical-specific capacitance parameters of this electrode are higher than pure WS-dAC, WS-dAC/NiO-0.001 and WS-dAC/NiO-0.01. SEM and TEM images of WS-dAC/NiO-0.005 are shown in Figure 1. Analysis of SEM images shows that mechanical activation of carbonizate leads to its grinding and contributes to a more uniform distribution of particle sizes of nanoporous carbon material obtained after the process of thermochemical activation of carbonizate. On the SEM image, magnified by 50,000 times, uniformly distributed nickel oxide nanoparticles are observed on the surface of individual particles. A deeper study of TEM analysis shows that nickel oxide nanoparticles have a spherical shape, and their diameter lies in the range of 30–50 nm.

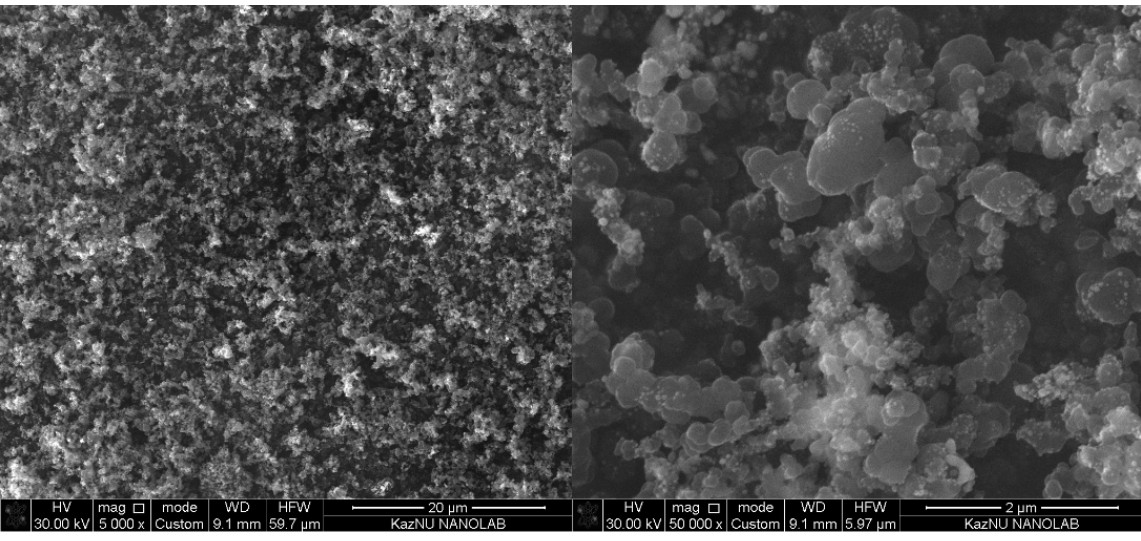

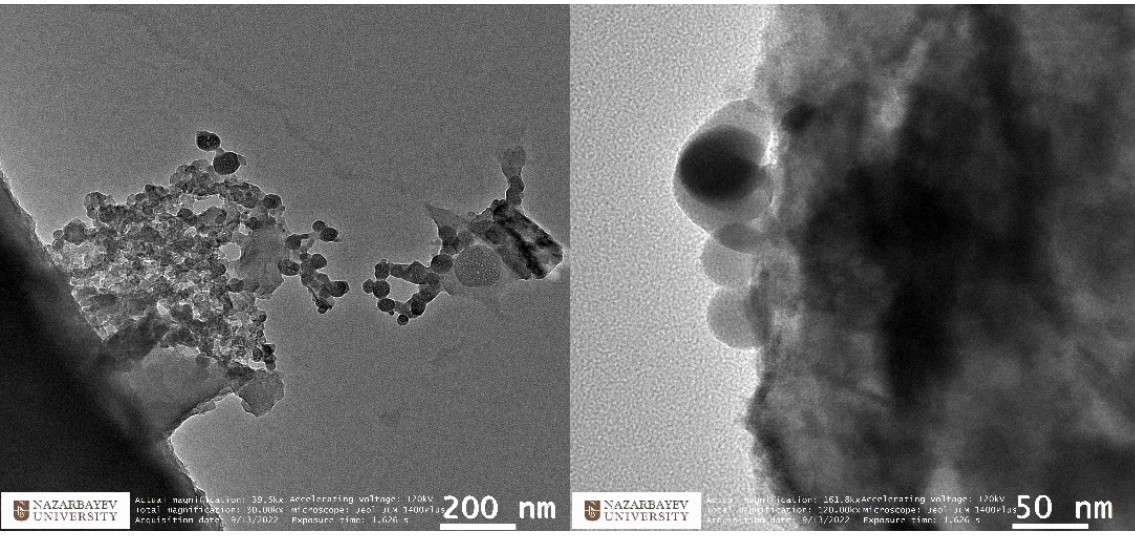

**Figure 1.** SEM and TEM images of WS-dAC/NiO-0.005.

Elemental content of the WS-dAC/NiO-0.005 sample was identified by Energy Dispersive X-ray Analysis (EDAX). Figure 2a–c shows the EDAX spectra of different parts of the sample surface, from which it can be seen that the sample consists mainly of carbon, oxygen, and nickel atoms. Potassium is the residue of a chemical activator (KOH). A small amount of Fe and Cr elements comes from the activation reactor.

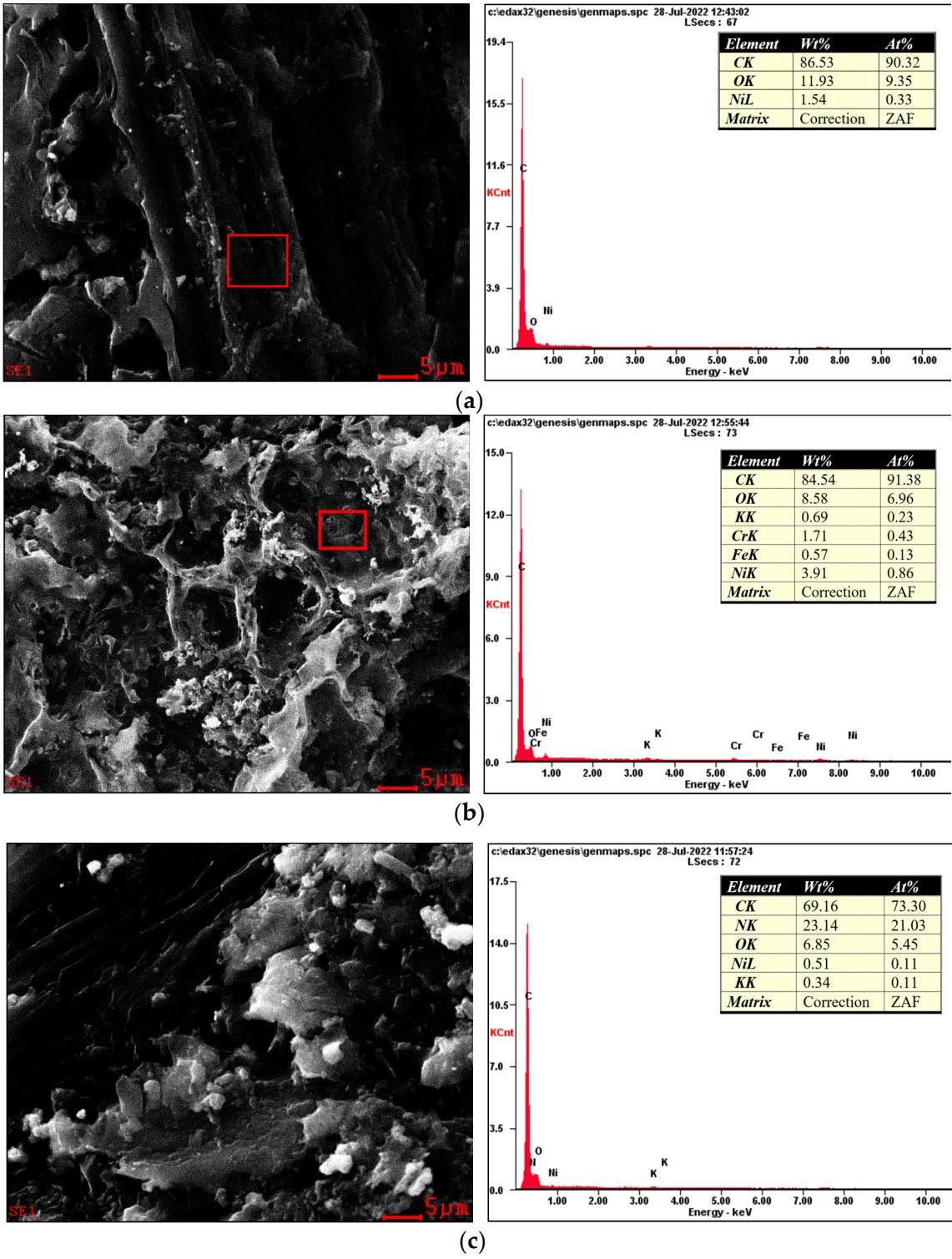

**Figure 2.** Scanning electron microscopy with energy-dispersive X-ray spectroscopy analysis results taken from different locations (red squares for figure (**a,b**) and the entire captured area for figure (**c**)) for the sample WS-dAC/NiO-0.005.

The results of a comparative analysis of the electrochemical characteristics of samples based on WS-dAC and its modification (WS-dAC/NiO-0.001, WS-dAC/NiO-0.005, WS-dAC/NiO-0.01) are shown in Figures 3–5. CV scanning was performed at sweep rates from 5 to 160 mV/s, and GCD was performed at a current density range of 100 to 10,000 mA/g.

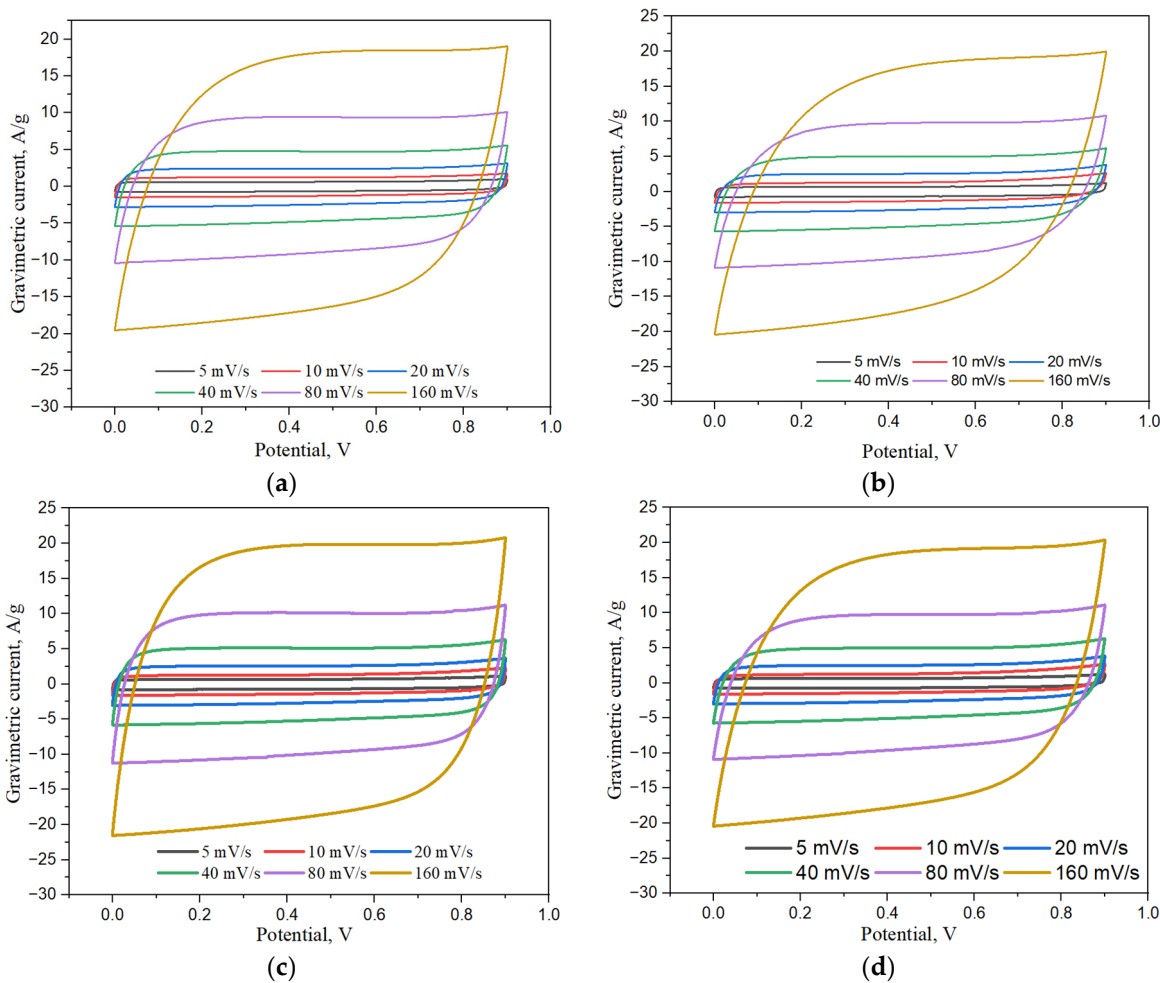

**Figure 3.** CV curves at different scan rates (5–160 mV/s) for (**a**) WS-dAC; (**b**) WS-dAC/NiO-0.001; (**c**) WS-dAC/NiO-0.005; (**d**) WS-dAC/NiO-0.01.

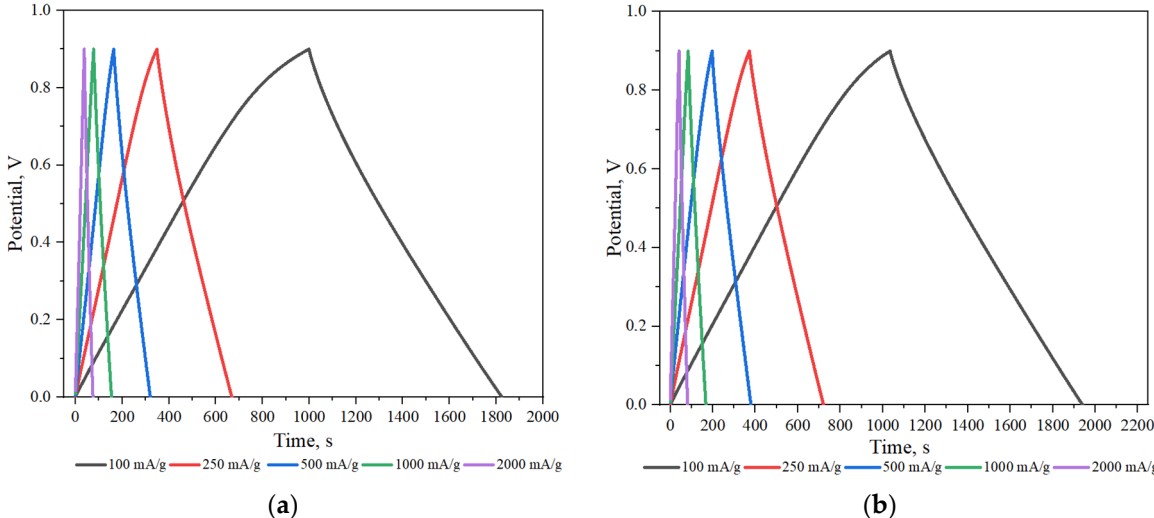

**Figure 4.** *Cont.*

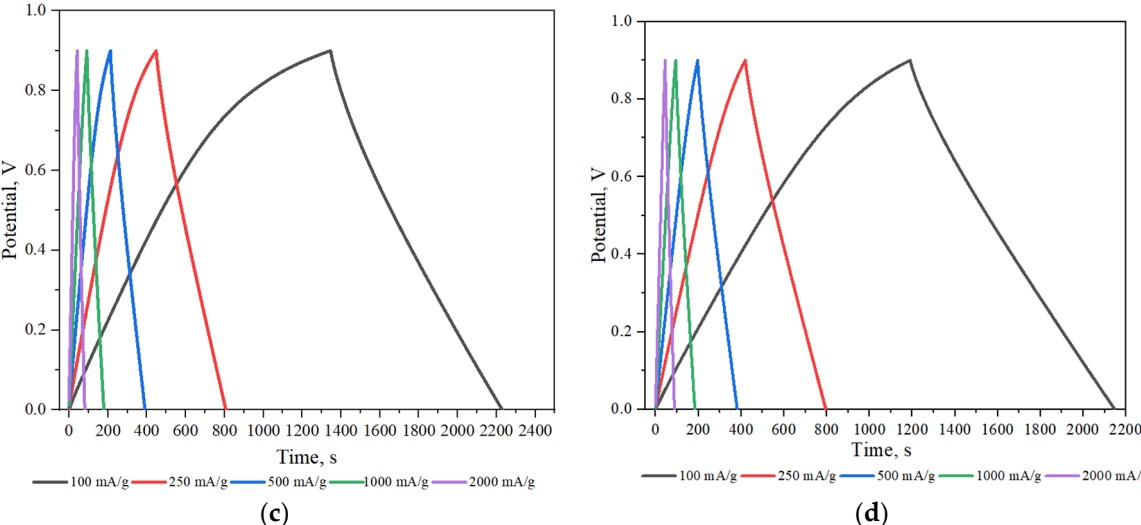

**Figure 4.** GCD curves at different charge–discharge current densities for (**a**) WS-dAC; (**b**) WS-dAC/NiO-0.001; (**c**) WS-dAC/NiO-0.005; (**d**) WS-dAC/NiO-0.01.

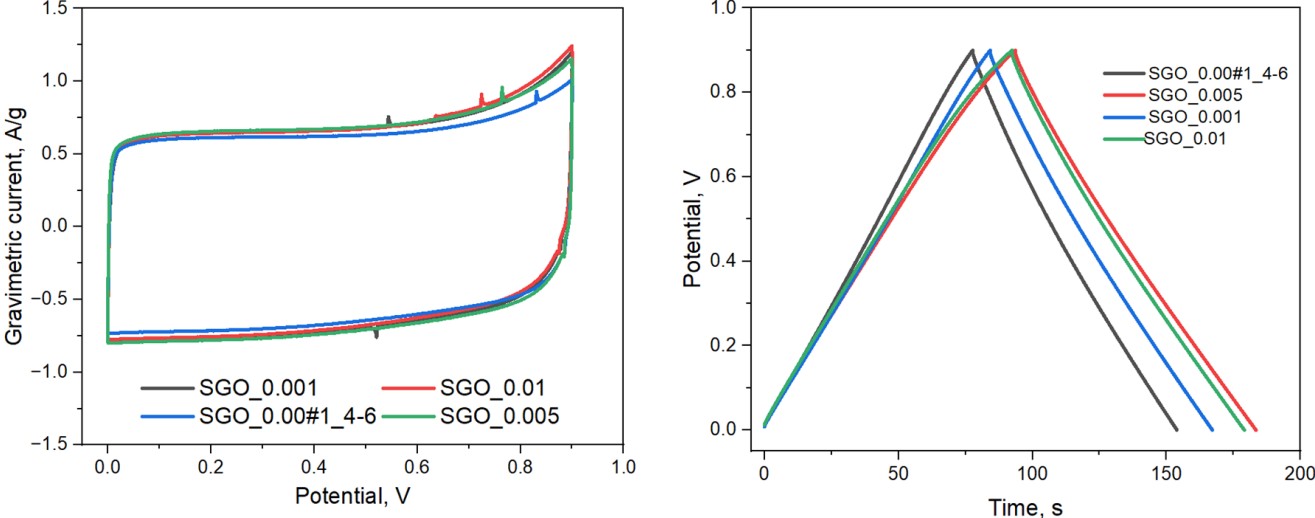

**Figure 5.** Electrochemical analysis of all samples measured in 6 M KOH; (**a**) Cyclic voltammetry curves at a potential scan rate of 5 mV/s; (**b**) Galvanostatic charge-discharge analysis curves at a current density of 1000 mA/g.

Figure 3 shows cyclic voltammetry curves at various CV sweep rates from 5 to 160 mV/s for WS-dAC and its modification (WS-dAC/NiO-0.001, WS-dAC/NiO-0.005, and WS-dAC/NiO-0.01). The voltammetric characteristics of the electrode samples are carried out in the potential range from 0.0–0.9 V. It can be seen from the CV curves that the fabricated cells based on the obtained materials operate as typical supercapacitors, as indicated by curves that have a similar geometry characteristic of supercapacitors with an electric double layer. The cyclic voltammetric curves of all samples taken for the two-electrode cell configuration did not give clear Faraday peaks. Firstly, this is due to the small amount of nickel oxide in the composition of the modified WS-dAC/NiO-x. Secondly, CV measurements were not carried out in the configuration with three electrodes. The capacitance values determined by the CV curve and the shape of the CV curves strongly depend on the configuration of the electrochemical cell [28,30]. The area of CV curves makes it possible to assume the specific capacitance values of the cells. The specific capacitance of WS-dAC/NiO-0.005 is higher than that of WS-dAC, WS-dAC/NiO-0.001, and WS-dAC/NiO-0.01 because the WS-dAC/NiO-0.005 sample has the largest rectangular surface

area under CV curve, than samples WS-dAC, WS-dAC/NiO-0.001, and WS-dAC/NiO-0.01. The best results were achieved with a WS-dAC/NiO-0.005 electrode reaching a specific capacitance of 279 F/g at a scan rate of 5 mV/s. This is an increase of almost 9% compared to pure WS-dAC electrodes, which have 254 F/g at the same scan rate.

## 4. Discussion

Figure 4 shows the characteristics of the galvanostatic charge-discharge in the potential range from 0 to 0.9 V for WS-dAC, WS-dAC/NiO-0.001, WS-dAC/NiO-0.005, and WS-dAC/NiO-0.01 samples at different current densities. Figure 4 shows that the electrode cells based on WS-dAC and its modification with nickel oxide nanoparticles WS-dAC/NiO-x operate as a supercapacitor with an electric double layer since the charge-discharge curves are symmetrical. The charge-discharge time for WS-dAC/NiO-0.005 electrodes is longer compared to WS-dAC, WS-dAC/NiO-0.001 and WS-dAC/NiO-0.01, which indicates a high specific capacitance of WS-dAC/NiO-0.005 electrodes, which was also visible on the CV curves (Figure 3). Due to the additional redox reaction, electrode cells based on WS-dAC/NiO-x composites have a higher capacitance compared to pure WS-dAC. At the same time, electrode cells based on WS-dAC/NiO-0.005 show an increase in capacity relative to pure WS-dAC by 16%. For electrodes based on unmodified pure WS-dAC at a gravimetric current density of 500 mA/g, the specific capacitance is 173 F/g. At the same current density for electrodes based on modified WS-dAC/NiO-0.005, the specific capacitance value is 205 F/g.

Figure 5 was presented for a visual comparison of the electrochemical performance of the samples. Figure 5a shows cyclic voltammetry curves of all the samples measured in electrolyte 6 M KOH at a potential scan rate of 5 mV/s. The CV curves reveal that the modified WS-dAC/NiO samples have a larger curve area than the WS-dAC electrodes, indicating that the samples based on modified activated walnut shell with nickel oxide have a relatively higher specific capacitance than unmodified WS-dAC electrodes. Figure 5b shows galvanostatic charge/discharge graphs of the pure activated walnut shell and its modified version with nickel oxide at a current density of 1000 mA/g and a potential range of 0 to 0.9 V. The charge-discharge time of the electrodes based on WS-dAC/NiO-0.005 is longer compared to other samples, which indicates a relatively high specific capacitance of WS-dAC/NiO-0.005.

It can be seen that the specific capacitance began to increase when the NiO content increased to a certain amount and then started to drop when higher amounts of nickel oxide were integrated into the activated walnut shell. This behavior may be due to lowering the EDLC capacity part because a large amount of NiO blocks the WS-dAC pores and NiO aggregation.

Figure 6a is a plot of capacitance versus current density for pure WS-dAC and its composites WS-dAC/NiO-x. The specific capacitances for electrodes based on pure WS-dAC and its composites WS-dAC/NiO-x were obtained from the GCD data according to [31]. Figure 6 shows that the specific capacitance of WS-dAC-based electrodes and its composites WS-dAC/NiO-x drops by an average of 21% with an increase in the charge-discharge current density from 100 to 10,000 mA/g (a hundredfold increase in current density), which indicates a good rate capability at high charge-discharge current densities.

The characteristic tendency of a decrease in specific capacitance with an increase in current density corresponds to supercapacitors and indicates that with an increase in current density, ions do not have time to fill the pore surfaces of individual sections due to the rate of the ion exchange reaction. Electrodes based on WS-dAC/NiO-x composites show a similar tendency to decrease in specific capacitance as pure WS-dAC, but it can be seen from the graph that the decrease in specific capacitance increases with an increase in the proportion of nickel oxide in the composition of WS-dAC/NiO-x composites. These higher capacitance losses with increasing current density compared to pure WS-dAC are characteristic of pseudo-capacitive supercapacitors [32].

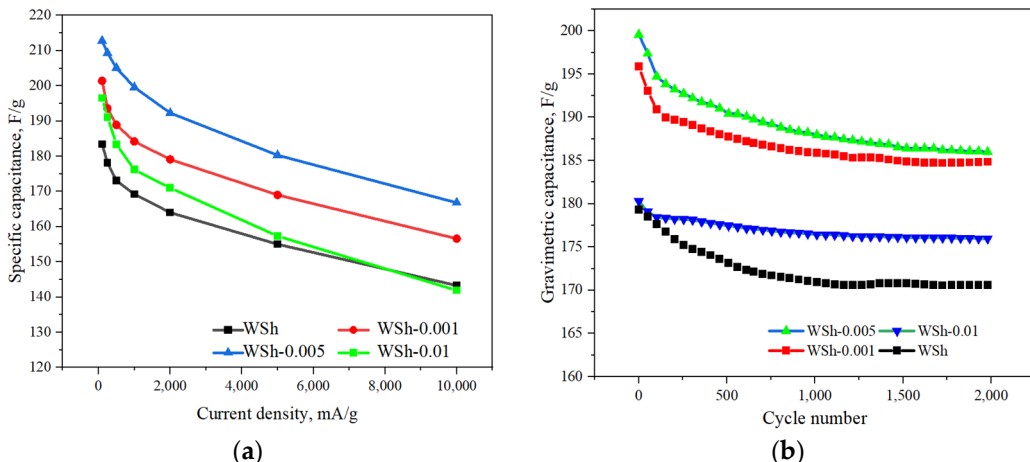

**Figure 6.** (**a**)—Specific capacitance of electrodes calculated from charge-discharge curves at various current densities; (**b**)—Specific capacitance versus physical cycle number performed at a constant current density of 1000 mA/g.

Figure 6b shows a plot of the gravimetric specific capacitance versus the number of physical cycles performed at a constant current density of 1000 mA/g. Supercapacitors based on WS-dAC/NiO-x composites show a sharp decline at the beginning of cycling and then demonstrate cyclic stability after 100 charge-discharge cycles in the potential range of 0–0.9 V. At the same time, Figure 6b shows that electrodes based on WS-dAC/NiO-0.01 are the most cyclically stable, maintaining ~98% specific capacity over 100 to 2000 test cycles.

## 5. Conclusions

A method for obtaining nanoporous carbon from WSs modified with NiO nanoparticles is proposed. It is shown that preliminary mechanical activation of carbonized material before the process of thermochemical activation contributes to a more uniform distribution of particle sizes of the obtained nanoporous carbon material. The preliminary mechanical activation on carbonized WSs in a high-energy 3D ball mill creates a highly developed porous structure in carbon materials with a specific surface area of 2200 ($\pm$50) m$^2$/g. Activated carbon from WSs was modified with spherical nickel oxide nanoparticles 30–50 nm in diameter by a single-stage thermochemical method using nickel nitrate hexahydrate Ni(NO$_3$)$_2$·6(H$_2$O) as a precursor. It has been established that the modification of activated carbon with nickel oxide nanoparticles makes it possible to increase the specific capacitance of the supercapacitor electrode by 16% compared to the original unmodified nanoporous carbon material.

**Author Contributions:** Conceptualization, B.L. and M.A.; methodology, M.Y. and N.R.; validation, A.T., Y.M. and G.U.; formal analysis, B.L. and M.Y.; investigation, A.T. and Y.M.; resources, M.A. and G.U.; data curation, N.R. and M.Y.; writing—original draft preparation, B.L. and M.A.; writing—review and editing, M.A. and M.Y.; visualization, M.A.; supervision, B.L.; project administration, B.L.; funding acquisition, B.L. All authors have read and agreed to the published version of the manuscript.

**Funding:** This research was funded by the Science Committee of the Ministry of Education and Science of the Republic of Kazakhstan ("Methods development for obtaining nanoporous materials with target properties from plant waste and creation of high-efficiency supercapacitors based on them", Grant No AP08856820).

**Data Availability Statement:** Not applicable.

**Conflicts of Interest:** The authors declare no conflict of interest.

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
