# Peer review of "Modification of Biomass-Derived Nanoporous Carbon with Nickel Oxide Nanoparticles for Supercapacitor Application"

_jcs, doi:10.3390/jcs7010020_

Round 1

Reviewer 1 Report

This is a good manuscript for the "Journal of Composites Science". It requires some minor modifications prior to acceptance. Please provide the EDXA analysis of more spots and present them in a chart. It is beneficial if the authors can provide some XRD patterns of the synthesized materials. Do authors have some information regarding the post-mortem analysis of the cells? It will help to increase the importance of the manuscript. The introduction needs expansion. The following references may help in this regard.

ChemEngineering 2020, 4(3), 43                                                                  

ACS Appl. Mater. Interfaces 2020, 12, 35, 39098–391

Author Response

Responses to Reviewer 1.

Thank you very much for your comprehensive reviewer.

(1) Please provide the EDXA analysis of more spots and present them in a chart

Author’s response: Thank you for your suggestion. We have added EDXA analysis results taken from different spots in the sample. (Figure 2. Results of Energy Dispersive X-ray Analysis (EDAX) for WS-dAC/NiO-0.005)

(2) Do authors have some information regarding the post-mortem analysis of the cells? It will help to increase the importance of the manuscript.

Author’s response: We did not perform a post-mortem analysis because we did not expect any structural changes after 2000 cycles. And we believe that electrodes based on carbon in KHO electrolyte can withstand many thousands of charge-discharge cycles.

(3) The introduction needs expansion. The following references may help in this regard.

Author’s response: We revised the introduction and made corrections according to reviewer comments. We reviewed the suggested articles and included one of them.

Reviewer 2 Report

did the author's check the XRD of samples? to confirm the purity of samples? 

authors should support the article with TG-DTA as some other papers prepared the walnut shell carbon in more than 500oC. 

Author Response

Responses to Reviewer 2.

  • Did the author's check the XRD of samples? to confirm the purity of samples? 

Author’s response:  XRD analysis of samples was carried out, but we did not include the XRD plots in the manuscript because they had uninformative broad and low-intensity peaks at around 25.876 for all samples. X-ray patterns of our materials are shown in the figure below. Our carbon material mainly contains a few-layer graphene structure according to X-ray diffraction and Raman spectroscopy analysis of our samples.

  • Authors should support the article with TG-DTA as some other papers prepared the walnut shell carbon in more than 500o

Author’s response:  In literature we found a lot of TG-DTA data of the walnut shell carbon but for this short time we did not have enough time to do this for our samples.

Reviewer 3 Report

The paper presented to me for review: " Modification of Biomass-Derived Nanoporous Carbon with 2 Nickel Oxide Nanoparticles for Supercapacitor Application" by Bakhytzhan Lesbayev et al. is written carefully. I accept this manuscript in present form.

Author Response

Dear editor,

We appreciate your acceptance of our paper for publication in the Journal of Composites Science.

Reviewer 4 Report

General Comments:

This manuscript reports the contribution of low-cost and eco-friendly biomass materials derived from Walnut shells in supercapacitors, the material was nano-porous carbonaceous material incorporated with nickel oxide (NiO) thermochemically. The author claims that the modification applied with nickel oxide nanoparticles results that the specific capacitance of the supercapacitor electrode was increased by 9% compared to the unmodified nano-porous carbon material. There has been literature about NiO/carbon material combo for the supercapacitor application, however, this work brings thermochemical processing of the biomass feedstock.

Overall, the manuscript is fine, it surely needs an extensive revision of English language and grammar and must be rewritten as there is a lot of repetition, the manuscript could be improved as at this current state, it is very difficult to even read through. It lacks the innovation and the novelty of this work, as NiO/Carbon combo (I repeat) have been published extensively for the applications of energy storage.

Specific comments:

The abstract can be improved, please start with a precise and brief statement of the topic, then provide the research purpose and methodology going further for what to expect from the finding and brief one-line conclusion. There is a lot of repetition in the first paragraph of introduction, the author should rearrange and try to communicate in rather straightforward way. Line 43, “The use of biomass waste as a feedstock for obtaining electrode materials for super-43 capacitors is discussed in [6–10].” Please mention at least in brief what those references talk about, instead of just giving a number. Same: Line 220, give a reference please.

The author mentioned thermochemical processing and then in experimental part, the author mentioned carbonization in a tube furnace, is the author referring to process of pyrolysis? I would rather explain a bit more the terminology author is using, as in the context of description, it gives another idea…. like mechanical activation, Etc. The experimental part with 2.1 Preparation of WS-dAC and 2.2 Modification of WS-dAC is written quite scruffy, I would suggest rewrite it in a manner, where less words can mean more. If possible, could the author please represent the parameters and conditions of the processes like CVD in a table for better understanding. Just a suggestion, if can please give names to these samples (WS-dAC, WS-dAC/NiO-0.001, WS-dAC/NiO-0.005, WS-dAC/NiO-0.01), to not to repeat all the time Again, please keep the terminology constant, for example in some cases authors writes KOH and in some potassium hydroxide. The sentence line 147-148 can be merged in the Materials characterizations, where author mentioned the same thing via SEM. Line 178, what’s modified ABC?

Did author investigate the adhesion of Nickle oxide or the agglomeration, what was the percentage of NiO in nanoporous carbon?

I suggest showing a comparative CV graph at 5mV for blank and modified WS-dAC/NiO, to see the difference clearly, specially to prove this point “This is an increase of almost 9% compared to pure WS-dAC electrodes which have 254 F/g at the same scan speed.” Same for the GCD curves. Perhaps, the author could provide an insight to why WS- dAC/NiO-0.01 has better cycling stability than the WS-243 dAC/NiO-0.005 that is proven to be the best composite regarding capacitance. Author mentioned Impedance measurements, but i do not see any. Author should try the lower current densities and scan rate for the samples WS- dAC/NiO-0.01 and WS-243 dAC/NiO-0.005 to prove the efficiency of the NiO and extra processes for the activation.

Please give a conclusion based on your insights and have a different aspect from the abstract of the manuscript.

Author Response

Responses to Reviewer 4.

Dear Reviewer, thanks a lot for your review, which indicates the weak parts of our article, this is helpful to improve its quality. The article was fully revised in accordance with the following comments:

  • Overall, the manuscript is fine, it surely needs an extensive revision of English language and grammar and must be rewritten as there is a lot of repetition, the manuscript could be improved as at this current state, it is very difficult to even read through.

Author’s response:  English and grammar were checked and revised throughout the article.

(2) “Line 43, “The use of biomass waste as a feedstock for obtaining electrode materials for super-43 capacitors is discussed in [6-10].” Please mention at least in brief what those references talk about, instead of just giving a number.” has been redone as follows:

Author’s response: “Application of biomass carbon electrode materials for the development of various types of supercapacitors with enhanced electrochemical performance, high-rate capability, and cycling stability is discussed in [7-11].”

(3) The experimental part with 2.1 Preparation of WS-dAC and 2.2 Modification of WS-dAC is written quite scruffy, I would suggest rewrite it in a manner, where less words can mean more. If possible, could the author please represent the parameters and conditions of the processes like CVD in a table for better understanding. Just a suggestion, if can please give names to these samples (WS-dAC, WS-dAC/NiO-0.001, WS-dAC/NiO-0.005, WS-dAC/NiO-0.01),

Author’s response: WS-dAC - walnut shells-derived activated carbon

WS-dAC/NiO-x - walnut shells-derived activated carbon modified with nickel oxide nanoparticles

WS-dAC/NiO-0.001 - walnut shells-derived activated carbon modified with nickel oxide nanoparticles in concentration 0.001

WS-dAC/NiO-0.005 - walnut shells-derived activated carbon modified with nickel oxide nanoparticles in concentration 0.005

WS-dAC/NiO-0.01- walnut shells-derived activated carbon modified with nickel oxide nanoparticles in concentration 0.01

(4) Please keep the terminology constant, for example in some cases authors writes KOH and in some potassium hydroxide.

Author’s response: Thank you for your suggestion. Now, we used only on the first sentence “The potassium hydroxide(KOH) was used as an activating agent” both potassium hydroxide and (KOH). For other sentences we used KOH.

(5) I suggest showing a comparative CV graph at 5mV for blank and modified WS-dAC/NiO, to see the difference clearly, specially to prove this point “This is an increase of almost 9% compared to pure WS-dAC electrodes which have 254 F/g at the same scan speed.” Same for the GCD curves. Perhaps, the author could provide an insight to why WS- dAC/NiO-0.01 has better cycling stability than the WS-243 dAC/NiO-0.005 that is proven to be the best composite regarding capacitance. Author should try the lower current densities and scan rate for the samples WS- dAC/NiO-0.01 and WS-243 dAC/NiO-0.005 to prove the efficiency of the NiO and extra processes for the activation.

Author’s response:  Thank you for pointing this out. We found your comments extremely helpful and have revised accordingly. Comparative CV curves were plotted for all samples at a scanning rate of 5mA/s. And also, comparative GCD curves were plotted at a current density of 1000mA/g for all samples.

(6) Figure 5 was presented for a visual comparison of the electrochemical performance of the samples. Figure 5a shows cyclic voltammetry curves of all the samples measured in electrolyte 6 M KOH at a potential scan rate of 5 mV/s. The CV curves reveal that the modified WS-dAC/NiO samples have a larger curve area than the WS-dAC electrodes, indicating that the samples based on modified activated walnut shell with nickel oxide have a relatively higher specific capacitance than unmodified WS-dAC electrodes. Fig. 5(b) shows galvanostatic charge/discharge graphs of the pure activated walnut shell and its modified version with nickel oxide at a current density of 1000mA/g and a potential range of 0 to 0.9 V. The charge-discharge time of the electrodes based on WS-dAC/NiO-0.005 is longer compared to other samples, which indicates a relatively high specific capacitance of WS-dAC/NiO-0.005.

Author’s response:  It can beed seen that the specific capacitance started to drop when higher amounts of nickel oxide were integrated to the activated walnut shell. This behavior may be due to lowering the EDLC capacity part because a large amount of NiO blocks the WS-dAC pores and NiO aggregation.

Round 2

Reviewer 1 Report

Revision is done appropriately.

Author Response

Thank you very much.

Reviewer 2 Report

its better to including a TGA measurement results or can support your  calcination temperature by reference

Author Response

Dear Reviewer, thank you a lot for your suggestion, this is helpful to improve quality of the manuscript. The article was fully revised in accordance with the following comment:

  • its better to including a TGA measurement results or can support your  calcination temperature by reference

Author’s response: We have included some results from previous studies regarding the thermal decomposition and kinetics of walnut shells during pyrolysis (the experimental part 2.1).

Reviewer 4 Report

The authors have done an effort but still I guess the comments were not cleared as I didn’t mean add more material, instead, I meant rephrase the introduction as its repetitive and a bit difficult to understand. The authors must be consistent through out the article, as in some cases they referred “in review [3]” in others “Ahmed et al.”. I insist please try to communicate in rather straightforward way and the article needs extensive change in writing and proof reading of English language. Experimental part and conclusion are much better but again please check the English and if needed rewrite for better understanding. I would say same for the abstract.

Below I would leave my previous comments and please try to improve the Article. Ignore the comments you have already improved.

Previous comments:

General Comments: This manuscript reports the contribution of low-cost and eco-friendly biomass materials derived from Walnut shells in supercapacitors, the material was nano-porous carbonaceous material incorporated with nickel oxide (NiO) thermochemically. The author claims that the modification applied with nickel oxide nanoparticles results that the specific capacitance of the supercapacitor electrode was increased by 9% compared to the unmodified nano-porous carbon material. There has been literature about NiO/carbon material combo for the supercapacitor application, however, this work brings thermochemical processing of the biomass feedstock.

Overall, the manuscript is fine, it surely needs an extensive revision of English language and grammar and must be rewritten as there is a lot of repetition, the manuscript could be improved as at this current state, it is very difficult to even read through. It lacks the innovation and the novelty of this work, as NiO/Carbon combo (I repeat) have been published extensively for the applications of energy storage.

Specific comments: The abstract can be improved, please start with a precise and brief statement of the topic, then provide the research purpose and methodology going further for what to expect from the finding and brief one-line conclusion. There is a lot of repetition in the first paragraph of introduction, the author should rearrange and try to communicate in rather straightforward way. Line 43, “The use of biomass waste as a feedstock for obtaining electrode materials for super-43 capacitors is discussed in [6–10].” Please mention at least in brief what those references talk about, instead of just giving a number. Same: Line 220, give a reference please.

The author mentioned thermochemical processing and then in experimental part, the author mentioned carbonization in a tube furnace, is the author referring to process of pyrolysis? I would rather explain a bit more the terminology author is using, as in the context of description, it gives another idea…. like mechanical activation, Etc. The experimental part with 2.1 Preparation of WS-dAC and 2.2 Modification of WS-dAC is written quite scruffy, I would suggest rewrite it in a manner, where less words can mean more. If possible, could the author please represent the parameters and conditions of the processes like CVD in a table for better understanding. Just a suggestion, if can please give names to these samples (WS-dAC, WS-dAC/NiO-0.001, WS-dAC/NiO-0.005, WS-dAC/NiO-0.01), to not to repeat all the time Again, please keep the terminology constant, for example in some cases authors writes KOH and in some potassium hydroxide. The sentence line 147-148 can be merged in the Materials characterizations, where author mentioned the same thing via SEM. Line 178, what’s modified ABC?

Did author investigate the adhesion of Nickle oxide or the agglomeration, what was the percentage of NiO in nanoporous carbon?

I suggest showing a comparative CV graph at 5mV for blank and modified WS-dAC/NiO, to see the difference clearly, specially to prove this point “This is an increase of almost 9% compared to pure WS-dAC electrodes which have 254 F/g at the same scan speed.” Same for the GCD curves. Perhaps, the author could provide an insight to why WS- dAC/NiO-0.01 has better cycling stability than the WS-243 dAC/NiO-0.005 that is proven to be the best composite regarding capacitance. Author mentioned Impedance measurements, but i do not see any. Author should try the lower current densities and scan rate for the samples WS- dAC/NiO-0.01 and WS-243 dAC/NiO-0.005 to prove the efficiency of the NiO and extra processes for the activation.

Please give a conclusion based on your insights and have a different aspect from the abstract of the manuscript.

Author Response

Dear Reviewer, thank you a lot for your suggestion, this is helpful to improve quality of the manuscript. The article was fully revised in accordance with the following comment:

The authors have done an effort but still I guess the comments were not cleared as I didn’t mean add more material, instead, I meant rephrase the introduction as its repetitive and a bit difficult to understand. The authors must be consistent through out the article, as in some cases they referred “in review [3]” in others “Ahmed et al.”. I insist please try to communicate in rather straightforward way and the article needs extensive change in writing and proof reading of English language. Experimental part and conclusion are much better but again please check the English and if needed rewrite for better understanding. I would say same for the abstract.

All of changes we noted yellow collars.

We have rewritten the abstract taking into account your remarks and shortened the introduction by removing some of the repetitive sentences.

We took your comments into account and tried to be consistent throughout the article (“Åžahin, M et al.”, “Ahmed et al.” etc. ).

Experimental part has been improved and expanded.

Round 3

Reviewer 4 Report

The manuscript have been improved, congratulations but please i repeat have someone look at English language and please remove unnecessary wordings.

Please correct the sentence Line 15-16.

Summarise the idea in 3 sentences from line 23-33. Also could you please rearrange the sentences from line 33-38.

The word also in line 41 may decline the importance of biomass importance, please rewrite the sentence to give a bit of credit to the facilities you are using due to their versatile properties.

Again, please revise and be constant à In the review 33 article [3] and Ahmed et al. [17]

I would also like if author could respond me clearly and seperately about the questions i asked:

I suggest showing a comparative CV graph at 5mV for blank and modified WS-dAC/NiO, to see the difference clearly, specially to prove this point “This is an increase of almost 9% compared to pure WS-dAC electrodes which have 254 F/g at the same scan speed.” Same for the GCD curves. Perhaps, the author could provide an insight to why WS- dAC/NiO-0.01 has better cycling stability than the WS-243 dAC/NiO-0.005 that is proven to be the best composite regarding capacitance. Author mentioned Impedance measurements, but i do not see any. Author should try the lower current densities and scan rate for the samples WS- dAC/NiO-0.01 and WS-243 dAC/NiO-0.005 to prove the efficiency of the NiO and extra processes for the activation.

Did author investigate the adhesion of Nickle oxide or the agglomeration, what was the percentage of NiO in nanoporous carbon?

Author Response

Responses to Reviewer

Dear Reviewer, thank you a lot for your suggestion, this is helpful to improve quality of the manuscript. The article was fully revised in accordance with the following comment:

  • Please correct the sentence Line 15-16.

Author’s response:

We have rewritten the abstract and this sentence.

  • Summarise the idea in 3 sentences from line 23-33. Also, could you please rearrange the sentences from line 33-38.

Author’s response:

We have rewritten the first paragraph and these sentences also.

  • The word also in line 41 may decline the importance of biomass importance, please rewrite the sentence to give a bit of credit to the facilities you are using due to their versatile properties.

Author’s response:

We added two sentences were indicated important of biomass-derived porous carbon materials

  • Again, please revise and be constantà In the review 33 article [3] and Ahmed et al. [17]

Author’s response:

We did it.

  • I would also like if author could respond me clearly and separately about the questions i asked:

I suggest showing a comparative CV graph at 5mV for blank and modified WS-dAC/NiO, to see the difference clearly, specially to prove this point “This is an increase of almost 9% compared to pure WS-dAC electrodes which have 254 F/g at the same scan speed.” Same for the GCD curves.

Author’s response:

Comparative CV curves were plotted for all samples at a scanning rate of 5mA/s. And also comparative GCD curves were plotted at a current density of 1000mA/g for all samples.

“ Figure 5 was presented for a visual comparison of the electrochemical performance of the samples. Figure 5a shows cyclic voltammetry curves of all the samples measured in electrolyte 6 M KOH at a potential scan rate of 5 mV/s. The CV curves reveal that the modified WS-dAC/NiO samples have a larger curve area than the WS-dAC electrodes, indicating that the samples based on modified activated walnut shell with nickel oxide have a relatively higher specific capacitance than unmodified WS-dAC electrodes. Fig. 5(b) shows galvanostatic charge/discharge graphs of the pure activated walnut shell and its modified version with nickel oxide at a current density of 1000mA/g and a potential range of 0 to 0.9 V. The charge-discharge time of the electrodes based on WS-dAC/NiO-0.005 is longer compared to other samples, which indicates a relatively high specific capacitance of WS-dAC/NiO-0.005.

  • Perhaps, the author could provide an insight to why WS- dAC/NiO-0.01 has better cycling stability than the WS-243 dAC/NiO-0.005 that is proven to be the best composite regarding capacitance.

Author’s response:

“It can beed seen that the specific capacitance began to increase when the NiO content increased to a certain amount and then started to drop when higher amounts of nickel oxide were integrated into the activated walnut shell. This behavior may be due to lowering the EDLC capacity part because a large amount of NiO blocks the WS-dAC pores and NiO aggregation.”

Initially, we expected that pure unmodified WS-dAC will show better cycling stabilty than NiO modified ones, however, sample with lowest NiO content (WS- dAC/NiO-0.01) was more stable under charge-discharge cycling. It is known that carbon electrodes are more stable than electrodes based on transition metal oxides and their derivatives. It can be assumed that the additional modification process (experimental part 2.2) not only helped to introduce nickel oxide nanoparticles into the carbon structure, but also changed the structure of carbon itself. Structural changes may be the reason why the WS-dAC/NiO-0.01 sample is more stable.

  • Author mentioned Impedance measurements, but i do not see any.

This is a typo; we removed that part of the sentence. We planned to make impedance measurements, but the electrochemical workstation for EIS analysis was broken.

  • Author should try the lower current densities and scan rate for the samples WS- dAC/NiO-0.01 and WS-243 dAC/NiO-0.005 to prove the efficiency of the NiO and extra processes for the activation.

Thank you for your suggestions. All the electrochemical experiments were performed with two electrode-configuration cell. We believe that the high sensitivity of three-electrode-configuration measurements can help understand the effect of NiO on carbon electrodes rather than the cell with the two-electrode configuration, even at lower current densities and scan rates. Some authors reported the dependence of capacitance values and the shape of the CV curves on the test cell configuration [https://doi.org/10.1002/ente.201300144, https://doi.org/10.1016/j.electacta.2004.10.078]. Unfortunately, we are unable to measure the cell with the 3-electrode configuration until the end of February next year due to our institute being closed for construction work.

Round 4

Reviewer 4 Report

I am satisfied from the answers from the Authors. I think the paper in its current form is ready but please do check one more time for mistakes and typos of English. I would encourage the Authors to try EIS studies and next time explain better the cell system.